# Socio-Spatial Segregation of Unhealthy Food Environments across Public Schools in Santiago, Chile

**DOI:** 10.3390/nu16010108

**Published:** 2023-12-28

**Authors:** Juliana Kain, Moisés H. Sandoval, Yasna Orellana, Natalie Cruz, Julia Díez, Gerardo Weisstaub

**Affiliations:** 1Institute of Nutrition and Food Technology (INTA), University of Chile, Macul 7830490, Chile; jkain@inta.uchile.cl (J.K.); yorellana@inta.uchile.cl (Y.O.); gweiss@inta.uchile.cl (G.W.); 2The Institute of Geography of the Pontifical Catholic University of Chile, Macul 7820436, Chile; nncruz@uc.cl; 3Public Health and Epidemiology Research Group, School of Medicine, Universidad de Alcala, 28801 Madrid, Spain; julia.diez@uah.es

**Keywords:** childhood obesity, food environment, unhealthy food outlet, Chile

## Abstract

Santiago, Chile is a very segregated city, with higher childhood obesity rates observed in vulnerable areas. We compared the counts and proximity of unhealthy food outlets (UFOs) around a 400 m buffer of 443 public schools (municipal and subsidized) located in socioeconomically diverse neighborhoods in 14 municipalities of Santiago. This was a cross-sectional study in which the socioeconomic status (SES) of the population living inside the buffer was classified as middle-high, middle, and low. We used the Kruskal–Wallis test for comparisons of density and proximity between type of school, SES, and population density. We used a negative binomial model (unadjusted and adjusted by population density) to determine the expected change in counts of UFOs by SES, which was compared to the reference (middle-high). Low SES neighborhoods had significantly more counts of UFOs, and these were located much closer to schools. Low and middle SES neighborhoods had an 88% and 48% higher relative risk of having UFOs compared to middle-high SES areas; (IRR = 1.88; 95% CI 1.59–2.23) and (IRR = 1.48; 95% CI 1.20–1.82), respectively. A socio-spatial segregation of UFOs associated with childhood obesity across public schools was observed in Santiago.

## 1. Introduction

Chile has a very high prevalence of childhood obesity. The latest figures obtained from the National Board for Student Aid and Scholarships (JUNAEB) of the Ministry of Education in 2022 [1] show that in 6- and 10-year-old school children these rates are 27.9% in both age groups. A unique feature that distinguishes this high rate from similar ones observed in other countries has been the rapid rise in prevalence, especially among more vulnerable children, which in the last 20 years has tripled [2].

Multiple studies in different societies have shown an association between socioeconomic status (SES) and obesity. For example, in the U.S.A, people who live in the poorest counties are more prone to obesity [3]. One of the reasons might be that in these counties violence is more prevalent, affecting physical activity, thus preventing people from being outdoors. Another study which was conducted in Toronto, Canada, demonstrated that, apart from low family income, children who live in highly deprived areas are more likely to be overweight [4].

The fact that the prevalence of childhood obesity follows a social gradient [5] has been reported not only in developed countries but also in middle-low- and low-income countries [6]. According to the World Bank, Chile is classified as a high-income country because per capita gross national income is around USD 13,000 (2020) [7] (slightly higher than the cut point); however, it is one of the most unequal societies in the world, as the income of the highest 20% income group is 10 times higher than the amount earned by the lowest quintile [8].

Studies on individual and environmental risk factors for childhood obesity have also shown that this condition is strongly influenced by other manifestations of poverty, such as low parental education, lack of proper access to health care, deficient neighborhood conditions, etc. [9]. Because these aspects are difficult to modify and/or it can take a very long time to observe improvements, we believe that in disadvantaged neighborhoods (if there are resources and political will) some environmental factors linked to food availability can be addressed in the short run and eventually lower the risk of childhood obesity. 

Research on the influence of food environments on obesity rates began many years ago; Glanz et al. [10] and Sallis et al. [11] being pioneers in this area. Many studies followed with inconsistent results, although most of them found an association with obesity and sedentary behavior. Recently, results from studies conducted in Beijing [12], Madrid [13], and The Hague [14], concluded that not only are there significantly more fast-food outlets around primary schools located in disadvantaged areas, but these are closer to schools than in better-off neighborhoods. 

On the other hand, a systematic review published in 2013 [15] that investigated the association between food outlets near schools with food purchases and consumption found “very little evidence of an association, but some effect on body weight”, as well as a more recent systematic review in 2020 [16] in which the authors analyzed 31 studies, finding a direct association between the density of unhealthy food outlets and obesity rates in only 14 of those studies. 

Despite these inconsistencies, most studies have shown that neighborhoods with the same sociodemographic characteristics, compared with those with good quality resources like parks, good street connectivity, better access to purchasing healthy foods, low crime rates, etc., have a lower obesity prevalence, although this is primarily because of increased physical activity. A recent study which analyzed the data of more than 20,000 US children from 1995 to 2021, with the aim of determining the association of neighborhood variables and social vulnerability during childhood, consistently showed that children in any of the following periods (early, middle, and late childhood) who lived in less vulnerable neighborhoods had a lower mean BMI trajectory from childhood to adolescence, independent of sociodemographic characteristics. Interestingly, the authors found that “vulnerability indices at birth were associated with the largest differences in mean BMI and subsequent obesity prevalence” [17].

Although most of these studies were conducted in the United States, Canada, some European countries, Australia, and New Zealand, recently some articles have been published that include Latin American countries. Zafra-Tanaka et al. [18], in a cross-sectional analysis of data pertaining to the social and built environments of 20,000 preschool children living in 159 large cities from six Latin American countries (Chile, Colombia, Guatemala, El Salvador, Mexico, and Peru), showed great variability in obesity prevalence, as well as in city and sub-city characteristics. The highest prevalence of excess weight was found in Chilean cities, while the lowest appeared in cities located in Colombia and Peru. The authors found that, in general, better living conditions were associated with a higher prevalence of excess weight (which is not the case in Chilean cities, as this paper will demonstrate), while the opposite was found with higher educational attainment. 

Based on this background information and the high childhood obesity rates observed among Chilean children, we considered it important to characterize the food environment in neighborhoods with different socioeconomic characteristics. Thus, the aim of this study was to compare density and proximity of unhealthy foods outlets around public schools located in socioeconomically diverse neighborhoods in the city of Santiago, Chile.

## 2. Materials and Methods

### 2.1. Setting and Sample 

In this observational cross-sectional study, we selected 14 municipalities in Santiago (out of a total of 52) based on two factors: their childhood obesity risk index, developed by Kain et al. [19], and our resources to conduct the study. We specifically included five municipalities with a high childhood obesity risk index, four with a medium risk, and five with a low risk. The mean obesity prevalence was 26.1%, 24.6%, and 19.1% respectively. These figures result from the average prevalence of obesity in children in first (6 year-olds) and fifth grades (10 year-olds) attending public schools in the 14 municipalities. The anthropometric data is collected nationally every year by the Ministry of Education and includes children attending those grades in either public schools that can be classified as municipal, or those which are free of charge and subsidized, in which parents pay a small monthly amount. Both types of schools include either primary education (kindergarten through eighth-grade) or primary plus secondary education. Around 35% of the national population of schoolchildren attend municipal schools, while 54% of children attend subsidized ones [20]. Municipal schools in general cater to lower-income children, while subsidized ones cater to both low- and middle-income children, depending on their location. 

It is important to point out that Santiago is a geographically segregated city, and that populations with higher SES tend to live in the outskirts of the city, especially in the northeastern side [21]. 

Our interest was to determine the density (counts) and location of every food outlet (400 m around every municipal and subsidized school of the 14 municipalities). Our units of analysis were the schools (municipal and subsidized); a total of 443 schools were included in this study.

### 2.2. Data Collection

Data was collected by eight observers between June and November 2022, and included both the type and location of every food store located within the 400 m buffer around the school. We chose this buffer as it is the one most reported in the literature [22]. Location was determined by an observer with the ArcGIS Survey123 app (version 3.18.145), which also allowed them to fill in a questionnaire, which in this case included the type of food outlet. These were categorized into convenience stores or minimarkets, supermarkets, bakeries (also sell cakes), restaurants (various types), fruit/vegetable stores and street carts (various types). The total area covered by was approximately 150 km^2^. 

The study was approved by the Ethics Committee of the Institute of Nutrition and Food Technology (INTA) of the University of Chile (Approval No 016/2022). 

### 2.3. Variables

Density or counts correspond to the total number of food stores within the 400 m buffer around the school, while unhealthy food outlets included minimarkets or convenience stores, fast food restaurants, bakeries, and street carts which sell unhealthy foods. 

Socioeconomic status (SES) refers to the SES of the population living inside the buffer zone. This index is based on the “Sociomaterial Territorial Index”, which is the only available data for small territorial units. This index is available per census block (based on the classification determined by the 2017 Population Census) and was elaborated considering the following: years of schooling of the head of household, level of crowding, and proportion of houses built with good, average, or low-quality materials [23]. This index, which can be considered as a proxy for SES, divides territories into percentiles using the same classification to define the SES of the Chilean population (high, middle-high, middle, middle-low, and low) [23,24] In this study, for simplicity, we considered three categories: middle-high, middle, and low SES. It is important to point out that this index reflects the characteristics of the population living around the school, and although in general children attend school in the same neighborhood where they live, we cannot assert that it represents the socioeconomic condition of the schoolchildren exactly. 

Population density: Because the shapes of these census units (census block) are irregular and do not coincide with the shape of the buffers, we created a geospatial tool with ArcPy on the ArcGIS Pro platform to add the proportion of persons for each buffer automatically. For example, if 30% of the area was inside the buffer, 30% of the population was added to the total population of the buffer. In addition, if a block was completely inside the buffer, its entire population was added to the total population of the buffer. In case schools were close to each other, their buffers and population overlapped. Then we divided the total population into quartiles. 

Concerning location: we determined the minimum distance (proximity) in meters from each school to the closest unhealthy food outlet. 

### 2.4. Statistical Analysis 

The counts of different types of unhealthy food stores are shown by school type and SES of the population living in the buffer zone. We calculated the median counts by school type, SES, and population density (in quartiles). Although we show counts with and without overlap, the analyses had to consider overlapping when schools were close to one another. We used the Kruskal–Wallis test for comparisons within each aspect and post hoc multiple comparisons to determine which were the categories of variables with different medians. All *p*-values below 0.05 are considered statistically significant.

We used a negative binomial model to examine the association between the SES of the population living inside the buffer and counts of unhealthy food stores. This model reports an incidence rate ratio (IRR) and confidence intervals (95% CI). In our study, the IRRs represent the expected change in the outcome measure (counts of unhealthy food outlets) by SES of the population compared to the reference group, that is middle-high. This model was calculated both unadjusted and adjusted by population density. When applying the binomial regression model, we performed a sensibility analysis removing food carts because these can vary both in distance and in operating hours around schools. However, the results (not presented here) did not differ significantly, so we decided to include them in the final adjustment. We performed all analyses using STATA/SE version 16.

## 3. Results

In the 14 studied municipalities, observers registered the data of the food environment around all public schools (*n* = 443). As Table 1 shows, 65.5% (*n* = 290) were subsidized schools and 35.5% (*n* = 153) municipal ones. Most of the population (71.6%) around these schools was classified as having a low SES, 15.6% as middle SES, and 12.9% as middle-high SES. Our estimates of population density within the 400 m surrounding the buffer resulted in 25% of the schools being in areas with a population composed of less than 3913 inhabitants, while the highest quartile included more than 7520. 

In total, observers identified 9939 unhealthy food outlets. As shown in Table 2, most unhealthy food outlets were convenience stores, followed by fast food restaurants and street carts. This table also shows that the distribution of food outlets with and without overlap is the same, so considering overlapping (which is the correct approach) can be regarded as having a low bias.

Table 3 presents the comparison between counts of unhealthy food outlets and their proximity according to type of school, SES, and population density. There is a significant association (*p*-value = 0.005) between counts and type of school. For example, in 25% of municipal schools there are a maximum of 28 unhealthy food outlets, while in 75% this figure reaches 53 establishments. On the other hand, in subsidized schools there are 22 and 48 premises, respectively.

Analyzing counts by SES, we observed a clear socioeconomic gradient (*p*-value < 0.001), that is, the higher SES, the fewer unhealthy food outlets. Table 3 also shows that in 25% of the schools located in areas with a population with middle-high SES, there are a maximum of six unhealthy food outlets, while in schools located in areas with low SES the maximum is 30. This same analysis was corroborated by performing a post-hoc test which showed that the median count of food outlets in the middle-high SES neighborhood is lower compared to the middle SES (*p*-value = 0.003), and this is significantly lower compared to the low SES (*p*-value < 0.001). Finally, as expected, counts are higher in neighborhoods with higher population densities (*p*-value < 0.001).

When analyzing the proximity of an unhealthy food outlet to the school by school type (right side of Table 3), we detected a significant difference (*p*-value = 0.050); on average, municipal schools have unhealthy food outlets closer compared to subsidized schools. Furthermore, in schools located in areas with low SES, proximity to unhealthy food stores was significantly lower than in schools located in neighborhoods with middle-high and middle SES (*p*-value = 0.010). For example, while in the most vulnerable sectors 50% of the unhealthy food stores are located 68 m from a school, in the middle-high SES areas the same percentage of stores are located 114.3 m away. Finally, as population density increases, the distance from unhealthy food outlets decreases (*p*-value < 0.001).

Figure 1 shows the spatial distribution of food outlets in the 14 municipalities according to the SES of the population living in the buffer around each municipal and subsidized school. This geographic distribution is clearly in line with the income inequality that exists in the city, with the northeastern part of the city (higher SES) having a significantly lower count of unhealthy food stores around schools.

To examine in greater detail the association between SES and unhealthy food outlets, we estimated a negative binomial regression model (Figure 2), with middle-high SES as the reference. This model is shown adjusted and unadjusted by population density. On average, schools located in the most vulnerable areas present an 88% higher relative risk of having unhealthy food stores, compared to middle-high SES areas (IRR = 1.88; 95% CI 1.59–2.23). Similarly, observing the socioeconomic gradient between schools, those located in areas with a population classified as middle SES have a 48% higher relative risk of higher counts compared to those with a middle-high SES (IRR = 1.48; 95% CI 1.20–1.82).

The variation of the IRR when adjusting by population density indicates that its inclusion produces an effect on the estimation of counts through more precise confidence intervals, that is, with a lower standard error.

## 4. Discussion

The main results of this study show that the density (counts) and proximity of unhealthy food outlets near public schools in Santiago follow a social gradient, contributing to the existing socio-spatial inequalities observed city-wide. In addition, these unhealthy food outlets are significantly closer to municipal rather than subsidized schools.

Previous research has also relied on cross-sectional analyses to explore food environments near schools—mostly due to the challenges of obtaining reliable and longitudinal data. Additionally, these authors have acknowledged the methodological challenges related, for example, to the categorization of food outlets, the size of the buffer around schools, and the most appropriate measure of accessibility [25,26]. These issues may partially explain the mixed evidence, as shown in a recent systematic review by Matsuzaki et al. [27] on the associations between food environment near schools and body weight by race/ethnicity, gender, grade, and socioeconomic status, which included 14 studies from the most developed countries. 

What is widely recognized is that in most countries, more-vulnerable population groups have less access to healthy food and a ubiquitous exposure to highly dense and nutrition-poor foods [28]. This, in turn, impacts their consumption patterns and poses higher risks of developing diet-related health issues [29]. This is particularly relevant during childhood, which is a very sensitive period for developing healthy behaviors [30].

As shown in this and many studies, not only is the number (counts) of unhealthy food outlets higher in disadvantaged areas, but also the distribution of the types of food stores is different according to the socio-economic characteristics of the neighborhood [31]. In our study there is a clear-cut difference in counts of convenience stores. In buffers where the population is more vulnerable, there are on average 28.6 convenience stores around each school compared to 19.2 and 10.7 in buffers around schools located in middle and high SES neighborhoods, respectively. A much smaller difference (or no difference) by SES was found for other types of food stores: fast food restaurants were between 6.4 and 5.7; bakeries were between 3 and 2.9; and unhealthy street carts were between 3.7 and 2.4.

Research on the association between convenience store access and childhood obesity has increased rapidly. A systematic review of 41 studies conducted in eight developed countries showed mixed results in the United States and the United Kingdom, while in studies from East Asia and Canada, these were generally non-significant. The authors remark on an important point which is the different definition that a convenience store may have depending on its location. As pointed out in this review, these types of stores may sell both healthy and unhealthy foods [32].

It seems that in urban settings in less developed countries, the number of convenience stores has a greater influence on childhood obesity (compared to other types of unhealthy food retail), as shown in a study conducted in two districts of Beijing, China, which included 2201 fourth grade students attending 37 primary schools. Both the nutritional health status of the children and the density of convenience stores were determined. The results showed that the association between convenience stores and childhood obesity was statistically significant [12]. In contrast, in more developed countries, such as the US, a larger clustering of fast-food restaurants around schools was reported to be associated with obesity in students; however, neighborhood SES was not considered [33,34].

In Latin America, similar results as were obtained in our study were reported in Costa Rica, Mexico, and Brazil. In Costa Rica, in a study that determined the nutritional status and lifestyle characteristics of 1268 children attending 10 public schools located in the Municipality of La Unión, there was a higher proportion of convenience stores around schools compared to other types of food outlets. Researchers also used a Geographic Information System (GIS) to locate food stores within a 400 m buffer around the schools. Of a total of 102 outlets, 25 corresponded to convenience stores and 22 to kiosks which mostly sell soft drinks and packaged foods. Schools with a higher mean prevalence of obesity were located in areas with a larger proportion of stores which sell unhealthy foods [35]. 

A different type of analysis on the same topic was conducted in Brazil in a study which compared the types of food sold (by processing level) in the surroundings of 30 private and 26 public elementary schools in the city of Niterói, Rio de Janeiro. Results showed that although the amount of ultra-processed foods sold around public and private schools was higher compared to other categories of foods, there was no difference between the amounts sold around public and private schools [36]. Also in Brazil, in 468 neighborhoods of Belo Horizonte [37] over a 10-year period, the establishment of unhealthy food outlets versus healthy ones was much higher in low-income neighborhoods. 

In Mexico, several studies have shown that in poor neighborhoods of Guadalajara, Puerto Vallarta, and Mexico City there is a significantly higher availability of retailers selling unhealthy foods [38]. In Lima, Peru, a study whose objective was to characterize the supply and advertising of foods and beverages inside 15 public and private schools, and street vendors surrounding the schools in 2019, found an unhealthy food environment both inside (which violates the law) as well as near the schools [39]. 

A study in Guatemala [40] mapped the food environment by SES in three neighborhoods (a high and middle SES urban neighborhood and a rural low SES one), considering a 150-m buffer around each school. Results showed that the highest concentration of unhealthy food outlets was found in middle-class urban and rural neighborhood. Because in our study we did not include rural communities, we can only state that our results are similar in that the highest SES had the lowest density of unhealthy food retail. 

Proximity to unhealthy food retail has also been addressed as a potential contributor to childhood obesity. For example, a study that included low-income households in New York City estimated the impact by grade level the distance of several types of retail had, finding that the greatest effects were observed among younger students (grades 3–8), where the incidence of obesity increased 1.4 percentage points, and the incidence of being overweight increased up to 1.9 percentage points, for every 0.1 mile closer to the nearest fast food outlet [41]. 

Another study examined the extent to which proximity to six types of food retail stores was associated with the nutritional health status of children attending 14 pediatric practices in eastern Massachusetts. The results showed that living closer to supermarkets and farther from fast-food restaurants was associated with a lower BMI z score, while convenience stores and fast-food restaurants had a stronger adverse effect on the BMI z score in lower-income neighborhoods [42]. 

In our study, there was a significant difference in the proximity to an unhealthy food outlet between neighborhoods; buffers with more-disadvantaged residents were 49 m closer to the school than in those that included middle-high SES populations. 

Interestingly, in low-income neighborhoods in New Jersey, USA [43], if the proportion of healthy foods sold by the same type of store (for example, grocery stores) around schools increased, the authors showed that this could influence the nutritional health status of children. This was shown around 33 public middle and high schools from four low-income communities. The authors reported that having a “small grocery store within 0.25 mile of school and an additional such store within that radius was associated with a lower BMI z-score while an additional supermarket within 0.25 mile of schools was associated with a lower probability of being overweight/obese”. Supermarkets which provide a large variety of foods are generally considered a healthy food retailer. A recent systematic review by Zhou et al. that included 24 studies (mainly from the US, Canada, and the UK) found a negative association between supermarket density and childhood obesity in half of the studies, a positive association in one fourth of them, and no significant differences in the rest [44]. 

In our study, counts of supermarkets were much higher in higher SES neighborhoods. The figures per 10,000 inhabitants in each SES category were 3, 0.7, and 0.54, respectively, with a significant difference between the extremes.

### Strengths and Limitations

In this study, a very large number of areas around public schools with different socioeconomic characteristics were observed, which gives assurance that these results are reliable and thus may be extrapolated to other municipalities in the country. 

Determining types of food stores remains a challenge and has been mostly done using business classifications, which are rarely validated against direct observation [45]; however, there are exceptions, as was shown in a study from the Netherlands in four urban and four rural neighborhoods [46] which concluded that commercially available data showed good to excellent statistical agreement (>0.71) with field audit data in urban and rural areas. To our knowledge, this validation study has not been conducted in Latin American cities. GIS classification has been shown to be reliable in counts but not in type of food store, as shown in Australia [47], where researchers compared GIS versus self-reported data on counts and types of food store, finding an association in counts but not in type of food store. In this study, the data was collected by trained observers who walked through the neighborhoods recording the data.

However, this study has some limitations. Even though the cross-sectional evidence for the association between unhealthy food establishments and childhood obesity has been shown to be significant, we do not know where people purchase their food. It has been shown that people with better SES levels not only have more nutrition knowledge but are able to travel farther to buy healthy foods, so maybe their local environment is not such a relevant issue for their dietary intake [29]. As previously stated, it is important to also determine people’s perception of the food environment (availability and prices) to move forward with further research [48].

Although the collection of high-quality data is very desirable, it must be recognized that fieldwork is costly, and this constitutes a limitation that makes this method not applicable to vast areas. Lastly, our study did not consider rural municipalities; that being said, only 15% of Chile’s population lives in rural areas, and there is no difference between the prevalence of childhood obesity between urban and rural areas [49]. 

## 5. Conclusions

In conclusion, there is a clear-cut socio-spatial segregation of unhealthy food environments associated with childhood obesity across public schools in Santiago, Chile. The results emphasize the need to address the quality of foods sold near schools, especially those located in vulnerable neighborhoods.

## Figures and Tables

**Figure 1 nutrients-16-00108-f001:**
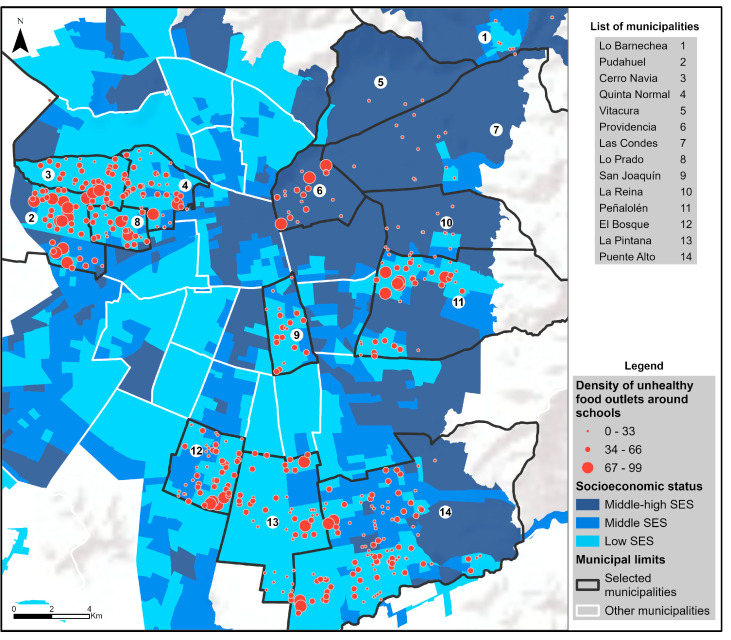
Density of unhealthy food outlets by neighborhood socioeconomic status for each of the 14 municipalities under observation in Santiago.

**Figure 2 nutrients-16-00108-f002:**
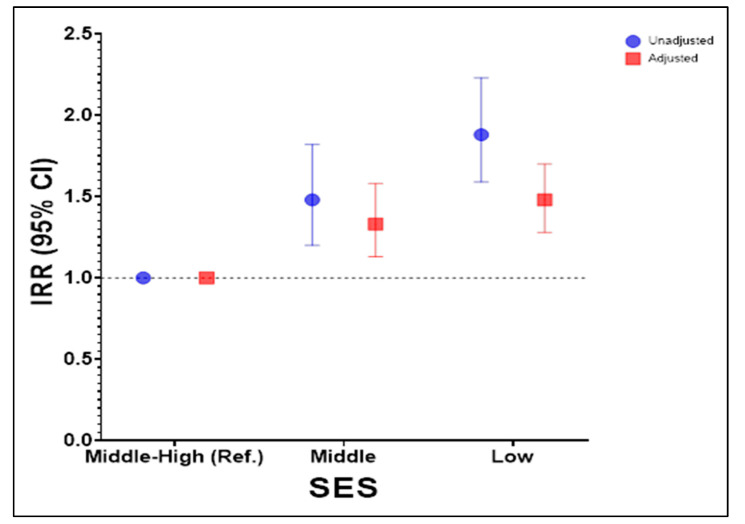
Negative binomial regression model of the number of unhealthy food outlets by SES of the population around the schools.

**Table 1 nutrients-16-00108-t001:** Descriptive characteristics of the study sample (*n* = 443 public schools).

Variables	*n* (%)	
Type of School		
Municipal	153	(34.54)
Subsidized	290	(65.46)
Area-level socioeconomic status (SES)		
Middle-High	57	(12.87)
Middle	69	(15.58)
Low	317	(71.56)
Population Density		
<3913	111	(25.06)
3914–5816	110	(24.83)
5817–7520	111	(25.06)
>7520	111	(25.06)

**Table 2 nutrients-16-00108-t002:** Distribution of counts of unhealthy food outlets in the 400 m buffer around schools, with and without overlapping.

Type of Food Outlet	No Overlapping		With Overlap	
	Absolute (N°)	Relative (%)	Absolute (N°)	Relative (%)
Minimarkets	6619	66.6%	10,894	66.4%
Fast food restaurants	1593	16.0%	2741	16.7%
Unhealthy street food carts	989	10.0%	1511	9.2%
Bakeries	738	7.4%	1269	7.7%
Total	9939	100.0%	16,415	100.0%

**Table 3 nutrients-16-00108-t003:** Comparison of counts of unhealthy food outlets and proximity by type of school, SES, and population density.

Variables	Density (Counts)					Proximity (Meters)			
	N ^1^	Median	p25	p75	*p*-Value	Median *	p25	p75	*p*-Value
Type of School					0.005				0.050
Municipal	6189	34 a	22	48		67.8	27.2	124.3	
Subsidized	10,226	40 b	28	53		74.25	45.7	131	
Socioeconomic Status					0.000				0.010
Middle-High	1238	16 a	6	34		114.3 b	40.6	167.2	
Middle	2218	30 b	20	42		69 a	31.2	123.4	
Low	12,959	41 c	30	53		68 a	31.05	119.35	
Population Density					0.000				0.000
<3913	2123	15 a	9	30		99.7 a	43.3	109.9	
3914–5816	3521	31 b	23	39		64.1 b	25.1	130.1	
5817–7520	4751	42 c	34	51		75.7 a	43.3	127.9	
>7520	6020	54 d	43	65		50.7 b	21.8	97.4	

^1^ Number of unhealthy food outlets with overlap. (*) Different letters indicate significant differences within each category and among the different categories. (*p*-value associated with the Kruskal–Wallis test).

## Data Availability

Data are unavailable due to privacy restrictions associated with the financing contract. The data may only be used by the research team.

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
