# Peer review of "Socio-Spatial Segregation of Unhealthy Food Environments across Public Schools in Santiago, Chile"

_nutrients, 2023, doi:10.3390/nu16010108_

Round 1

Reviewer 1 Report

Comments and Suggestions for Authors

Dear Authors:

Regarding the manuscript with title “Socio-Spatial Segregation of Unhealthy Food Environments across Public Schools in Santiago, Chile”, I have some minor comments to address.

Comment 1:

Line 14: Authors must clarify the meaning of “SES”, as i tis the first time they refer to it.

Comment 2:

On line 15, authors refer that SES was classified as “middle-high, medium, and low”. On Results (line 20), authors refer to middle-low neighborhoods.

Comment 3:

Line 18-19: Authors must delete this sentence “66% of the UFO corresponded to minimarkets”, as i tis only a descriptive data of the study and not na answer to the aim of the study.

Comment 4:

Line 27: ”show that in 6- and 10-year-old school children, these rates are 27.9 % in both age groups”. What are the ages of the children/adolescents that integrate this study? The term childhood, used during the manuscript, include all ages?

Comment 5:

Line 71: Authors must change “Europe” by “countries from Europe”

Comment 6:

Line 86: Regarding what authors presente on Results, I suggest to add a secondary purpose of the study related to the comparison of the density and proximity of unhealthy food outlets around public schools according to type of school and population density. The primary purpose of ths study remains: to compare the density and proximity of unhealthy food outlets around public schools according to socioeconomic status.

Comment 7:

Why authors presente results from 14 of 52 municipalities of Santiago, Chile?

Comment 8:

Line 114: Authors must change “as it the one” by “as it is the one”

Comment 9:

Line 225: Why authors considered minimarkets as a store that sells unhealthy food and not supermarkets?

Comment 10:

Lines 131-132: “This index which can be considered as a proxy for SES divides territories into quintiles”. I please ask authors to specify these five categories. How these 5 categories were converted to three categories?

Comment 11:

Lines 146-147: Authors must change “Finally, we determined the minimum distance (proximity) in meters to the closest unhealthy food outlet.” by “In what concerns location, it was determined the minimum distance (proximity) in meters from each school to the closest unhealthy food outlet.”

Comment 12:

On Statistical analysis authors have to add a sentence refering to which p value differences were considered statistically significant?

Comment 13:

Line 170: Authors must change “Most of the population” by “Most of the population (71.6%)”.

Comment 14:

Lines 172-174: “Our estimates of population density within the 400 m surrounding the buffer resulted in 25% of the schools being in areas with a low density, half in medium-dense areas, and 25% in areas with a high density.” In which reference authors based this classification? (low density, medium dense áreas and high density?)

Comment 15:

Line 177: Authors must change “9,939 food outlets” by “9,939 unhealthy food outlets”.

Comment 16:

On Table 2, I suggest authors to presente data from the highest to the lowest count.

Comment 17:

On Table 3:

1.       authors must refer the difference between letters “a”, “b” and “c”.

2.       On legend what means 1 Schools with overlap?

3.       Authors must present always the comma as “.” and not as “,”

4.       The “n” represented on Table refers to the number of public schools regarding each variable or what it means?

Comment 18:

Line 237: Authors must change “density (counts)” by “density (counts) and proximity”

Comment 19:

If authors add the secondary objective, they have to add a secondary Conclusion

Comments on the Quality of English Language

Minor editing of English required. Some minor comments are addressed to that point.

Author Response

Thank you very much for your useful comments and pointing out our mistakes on the proper use of English.

Please find the answers below:

      1, 2 and 3. These refer to the Abstract and they have been addressed. Due to the Word limit of 200, we deleted the phrase “These have been shown to have a poor environment.”

  1. Obesity rates in the study were obtained from an average of these rates in 1st and 5th grades which are part of the grades with anthropometric data collected by the Ministry of Education. In the article it appears like this

These figures result from the average prevalence of obesity in children in first (6-year-olds) and fifth grades (10-year-olds) attending public schools in the 14 municipalities. The anthropometric data is collected every year nationwide by the Ministry of Education

  1. We included “some European countries.”
  2. Thank you for this observation, however the purpose of this paper is to show the unequal distribution of unhealthy food stores by SES in terms of counts and distance to schools.
  3. We now explain why the sample included 14 municipalities as follows:

In this observational cross-sectional study, we selected 14 municipalities in Santiago (out of a total of 52) based on two factors: their childhood obesity risk index developed by Kain et al [17] and our resources to conduct the study. 

  1. We made the change.
  2. Minimarkets are convenience stores with almost no healthy products. What we observed was that about 1/3 of them had outside the store a couple of low-quality fruits (mostly apples and bananas) and vegetables (carrots and onions) in contrast to supermarkets which have a good supply of healthy foods.
  3. We now explain the 5 categories of SES. If we would have considered 5 categories, the N of each was much smaller and the interpretation of the main results more difficult.
  4. We made the suggested change
  5. We now included the p value
  6. We made the suggested change
  7. Regarding population density, we now modified the words low, medium and high density, because we can only say that we divided density into quartiles. It is now written as follows:

Our estimates of population density within the 400 m surrounding the buffer resulted in 25% of the schools being in areas with a population composed of less than 3,913 inhabitants, while the highest quartile included more than 7,520.

  1. We made the suggested change
  2. We inserted a new Table 2 presenting the data as suggested by the Reviewer
  3. We now explain what the N stands for and what the different letters mean as follows:
  • Different letters indicate significant differences within each category and among the different categories. (p-value associated with the Kruskal Wallis test).
  1. We made the change
  2. As explained before, we only included in this article one main objective

Thank you again for helping us in improving the editing of the English language. 

Reviewer 2 Report

Comments and Suggestions for Authors

In this manuscript, the authors compared about the density and proximity between type of school, SES and population density and found that low SES neighborhoods had more counts of unhealthy food outlets and the distance from school to these markets was much closer, which indicate that a socio-spatial segregation of UFO associated with childhood obesity across public schools is observed in Santiago, Chile. The whole manuscript is well organized and written, however, there are still some comments need to be addressed as followed:

1.    Something is missing in the end of line 157.

2.    The quality of the figure 1 should be improved.

3.    It would be better if the author state Socioeconomic Status (SES) in the beginning of the manuscript.

Comments on the Quality of English Language

In this manuscript, the authors compared about the density and proximity between type of school, SES and population density and found that low SES neighborhoods had more counts of unhealthy food outlets and the distance from school to these markets was much closer, which indicate that a socio-spatial segregation of UFO associated with childhood obesity across public schools is observed in Santiago, Chile. The whole manuscript is well organized and written, however, there are still some comments need to be addressed as followed:

1.    Something is missing in the end of line 157.

2.    The quality of the figure 1 should be improved.

3.    It would be better if the author state Socioeconomic Status (SES) in the beginning of the manuscript.

Author Response

Thank you very much for your comments which were all addressed.

  1. We completed the sentence in line 157.
  2. Figure 1 has been improved.
  3. We now included the origin of our definition of SES in the Abstract and a paragraph in the Introduction introducing the association between poverty and childhood obesity.

Multiple studies have shown the association between Socioeconomic Status (SES) and obesity. For example, in the U.S.A, people who live in the poorest counties, are more prone to obesity. One of the reasons might be that in these counties, violence is more prevalent affecting physical activity, thus, preventing people to be outdoors. Another study which was conducted in Toronto, Canada, demonstrated that apart from low family income, children who live in highly deprived areas are more likely to be overweight.